# A genome-wide association study implicates the olfactory system in *Drosophila melanogaster* diapause-associated lifespan extension and fecundity

Sreesankar Easwaran, Denise J Montell*

Molecular, Cellular, and Developmental Biology Department University of California, Santa Barbara, Santa Barbara, United States

## eLife Assessment

This **important** study shows how genetic variation is associated with fecundity following a period of reproductive diapause in female *Drosophila*. The work identifies the olfactory system as central to successful diapause with associated changes in longevity and fecundity. While the methods used are **convincing**, a limitation of the study, as of any other laboratory-based investigation is the challenge of demonstrating how well measures for fitness related to diapause and its recovery correlates with realities encountered during development in the wild.

*For correspondence:
denise.montell@lifesci.ucsb.edu

Competing interest: The authors declare that no competing interests exist.

## Abstract

The effects of environmental stress on animal life are gaining importance with climate change. Diapause is a dormancy program that occurs in response to an adverse environment, followed by resumption of development and reproduction upon the return of favorable conditions. Diapause is a complex trait, so we leveraged the Drosophila Genetic Reference Panel (DGRP) lines and conducted a genome-wide association study (GWAS) to characterize the genetic basis of diapause. We assessed post-diapause and non-diapause fecundity across 193 DGRP lines. GWAS revealed 546 genetic variants, encompassing single nucleotide polymorphisms, insertions, and deletions associated with post-diapause fecundity. We identified 291 candidate diapause-associated genes, 40 of which had previously been associated with diapause, and 89 of which were associated with more than one SNP. Gene network analysis indicated that the diapause-associated genes were primarily linked to neuronal and reproductive system development. Similarly, comparison with results from other fly GWAS revealed the greatest overlap with olfactory-behavior-associated and fecundity-and-lifespan-associated genes. An RNAi screen of selected candidates identified two neuronal genes, Dip-γ and Scribbler, to be required during recovery for post-diapause fecundity. We complemented the genetic analysis with a test of which neurons are required for successful diapause. We found that although amputation of the antenna had little to no effect on non-diapause lifespan, it reduced diapause lifespan and post-diapause fecundity. We further show that olfactory receptor neurons and temperature-sensing neurons are required for successful recovery from diapause. Our results provide insights into the molecular, cellular, and genetic basis of adult reproductive diapause in *Drosophila*.

## Introduction

To endure adverse environments, many organisms from worms and flies to some mammals initiate a dormancy program called diapause, triggered by cues like unfavorable temperatures, drought, or starvation (*Deng et al., 2018*; *Easwaran and Montell, 2023*; *Hand et al., 2016*; *Hussein et al., 2022*; *Tatar and Yin, 2001*). By enhancing survival, diapause can support both beneficial insects like pollinators and harmful insects such as agricultural pests and disease vectors. Elucidating diapause mechanisms may suggest ways to control harmful insects and support beneficial organisms. Furthermore, unraveling diapause genetics holds promise for understanding stem cell resilience (*Easwaran et al., 2022*), aging (*Fontana et al., 2010*; *Hutfilz, 2022*), and even cancer dormancy (*Easwaran and Montell, 2023*; *Lin and Zhu, 2021*).

Successful diapause occurs optimally at different developmental stages depending on the organism (*Denlinger, 2002*; *Hahn and Denlinger, 2011*; *Kostál, 2006*). In *Drosophila* species, the optimal stage is the newly eclosed adult. During diapause, feeding, activity levels, and mating behaviors change, metabolism slows, reproduction arrests, stress resilience increases, and lifespan is lengthened (*Hutfilz, 2022*; *Kubrak et al., 2016*; *Kubrak et al., 2014*; *Reis et al., 2015*; *Tatar and Yin, 2001*; *Tatar et al., 2001*; *Zonato et al., 2017*).

The genetic basis of adult reproductive diapause in *Drosophila* is likely complex because the diapause program involves sensing environmental cues and responding by reprogramming behavior, metabolism, reproduction, and aging, continuously monitoring environmental conditions, and reactivating development and/or reproduction when conditions become favorable. Historically, studies of *Drosophila* diapause have focused primarily on entry into diapause by measuring arrest of egg production, specifically at the stage of yolk accumulation referred to as vitellogenesis (*Saunders et al., 1990*; *Saunders et al., 1989*). However, it has become clear that diapause is a comprehensive program that changes virtually every aspect of the fly's life from behavior to metabolism, reproduction, and lifespan (*Denlinger, 2023*; *Kubrak et al., 2014*; *Tatar and Yin, 2001*). Prior studies have identified temperature and day length as relevant environmental cues and key hormones such as insulin, juvenile hormone (JH), and 20-hydroxyecdysone, which coordinate metabolism and reproduction (*Denlinger, 2023*; *Flatt et al., 2008*; *Hutfilz, 2022*; *Saunders et al., 1990*; *Sim and Denlinger, 2013*; *Tatar and Yin, 2001*; *Tatar, 2004*; *Tatar et al., 2001*; *Williams et al., 2006*). Recent studies identify specific circadian neurons that sense changes in temperature and relay that information to the reproductive system (*Hidalgo et al., 2023*; *Meiselman et al., 2022*). Yet, much remains to be learned about this fascinating life history trait (*Denlinger, 2023*).

One approach to analyzing complex traits and mapping quantitative trait loci is to conduct a genome-wide association study (GWAS) (*Huang et al., 2014*; *Morozova et al., 2015*; *Shorter et al., 2015*). The Drosophila Genetic Reference Panel (DGRP) comprises fully sequenced, highly inbred lines of *Drosophila melanogaster* and serves as a valuable tool for understanding genotype-phenotype relationships in *Drosophila* (*Mackay and Huang, 2018*; *Morozova et al., 2015*).

We used the DGRP to carry out a *Drosophila* GWAS and identify genes and gene classes associated with successful diapause. We developed a novel method for assessing successful diapause, by evaluating not only the arrest of egg development or the recovery of egg maturation but a more stringent criterion: the ability of post-diapause flies to produce viable adult progeny. Of the 291 diapause-associated genes we identified, ~40 had previously been implicated in diapause in flies or another organism. The classes of genes most overrepresented in the diapause-associated set are genes that regulate neuronal development and/or gonad development. We show that *Gal4*-mediated RNAi knockdown is effective at 10°C, and we identify two neuronal genes required during recovery for post-diapause fecundity. We further complement this genetic analysis by identifying neurons in the fly antenna required for successful diapause. Our studies implicate the olfactory system as critically important in successful diapause.

## Results

### Quantifying successful diapause recovery across the DGRP lines

Key features of *Drosophila* diapause include ovarian arrest and post-diapause recovery of fertility (the ability to produce any progeny) and fecundity (the number of progeny produced). Whereas the majority of studies of *Drosophila* diapause focus on arrest of oogenesis, we chose to quantify diapause

recovery by assessing the ability of newly eclosed flies to undergo 35 days of diapause, recover, and produce viable progeny, which is a more stringent test of success (*Figure 1A*). Additionally, in pilot experiments, post-diapause fecundity exhibited sufficient variation to be useful for a GWAS.

DGRP lines (n=193) were allowed to develop under non-diapause conditions (25°C and 12:12 L:D). Virgin females were collected and transferred to diapausing conditions (10°C and 8:16 L:D) for 5 weeks. Subsequently, flies were shifted to 18°C for 1 day, followed by incubation at 25°C in a fresh vial of fly food for recovery. We chose 5 weeks based on pilot studies that showed that nearly all DGRP lines showed excellent survival at 5 weeks in diapause conditions while exhibiting sufficient variation in post-diapause fecundity to carry out GWAS. Beyond 5 weeks, fecundity was low, and there was insufficient variation for a GWAS.

We evaluated the ability of post-diapause flies to reproduce by mating individual females with two Canton-S (CS) male control (non-diapausing) flies for 4 days. Afterward, parents were removed, and progeny were allowed to develop for 12 more days. We then measured post-diapause fecundity by counting the number of adult progeny that eclosed. A minimum of 20 female flies per DGRP line were tested. Non-diapause fecundity was measured by setting crosses immediately upon collecting virgin flies. The average non-diapause and post-diapause 4-day fecundity of DGRP lines is shown in *Figure 1B*, arranged in ascending order of non-diapause fecundity. In general, most lines showed lower fecundity post-diapause compared to newly eclosed flies in optimal conditions, as previously reported for one inbred wild strain (*Tatar et al., 2001*). However, there were interesting exceptions. For example, lines 309, 738, and 853 exhibited better fecundity post-diapause compared even to young flies in optimal conditions (*Figure 1—source data 1*). The DGRP lines exhibit variability in fecundity both post-diapause and in non-diapause conditions (*Figure 1B*; *Durham et al., 2014*). Therefore, we normalized post-diapause to non-diapause fecundity (*Figure 1C*, *Figure 1—source data 1*). The broad-sense heritability for normalized post-diapause fecundity was 0.51 (see Materials and methods). The lines that showed poor non-diapause fecundity that improved after diapause (e.g. 362, 737, 822) suggested that there might be a genetic trade-off between the two. However, the overall correlation (*Figure 1D*) between non-diapause and post-diapause fecundity was 32.3% ($R^2$=0.3228, Pearson's correlation coefficient, r=0.5682), suggesting a positive rather than negative correlation overall between post-diapause and non-diapause fecundity. The frequency distributions of 4-day fecundity for the non-diapause (*Figure 1E*) and normalized fecundity (*Figure 1F*) conditions are shown. Although fecundity is lower post-diapause than in young flies, even in CS controls, it is higher than for flies maintained in non-diapause conditions for 35 or 42 days (*Figure 1G*, *Figure 1—source data 3*).

## GWAS for diapause using the DGRP tool

The normalized fecundity scores (average of normalized post-diapause fecundity [individual post-diapause 4-day fecundity/average non-diapause 4-day fecundity]) served as the basis for conducting a GWAS using the DGRP2 web tool (*Huang et al., 2014*; *Mackay et al., 2012*). A total of 546 genetic variants, encompassing single nucleotide polymorphisms (SNPs) and insertions/deletions, were identified as associated with diapause fecundity (*Figure 2A and B*, *Figure 2—source data 1*). When a variant is located within 1 kb up- or downstream of an annotated gene, or within the gene, it is considered potentially associated with that gene. We thus identified 291 candidate diapause-associated fecundity genes. Notably, 40 out of the 291 genes had previously been reported to be associated with diapause (*Figure 2—source data 2*) either through functional analysis (e.g. insulin receptor and *couch potato* [*cpo*]) (*Kankare et al., 2012*; *Kubrak et al., 2014*; *Schmidt et al., 2008*; *Sim and Denlinger, 2013*; *Zhang and Denlinger, 2011*), due to changes in gene expression (e.g. expanded [*ex*] and Laminin A [*LanA*]) (*Zhao et al., 2016*), or associated with cold tolerance (β-Tub97EF; *Myachina et al., 2017*; *Figure 2—source data 2*). Eighty-nine genes were associated with more than one diapause-associated variant (e.g. at least two different SNPs) (*Figure 2C*).

## Network analysis of diapause-associated genes

Using the gene list obtained from the GWAS, we performed a gene network analysis using the GeneMANIA application within Cytoscape (*Shannon et al., 2003*; *Warde-Farley et al., 2010*). This application facilitates the generation of network predictions based on a combination of known physical and genetic interactions, co-localization, co-expression, shared protein domains, pathway data, and

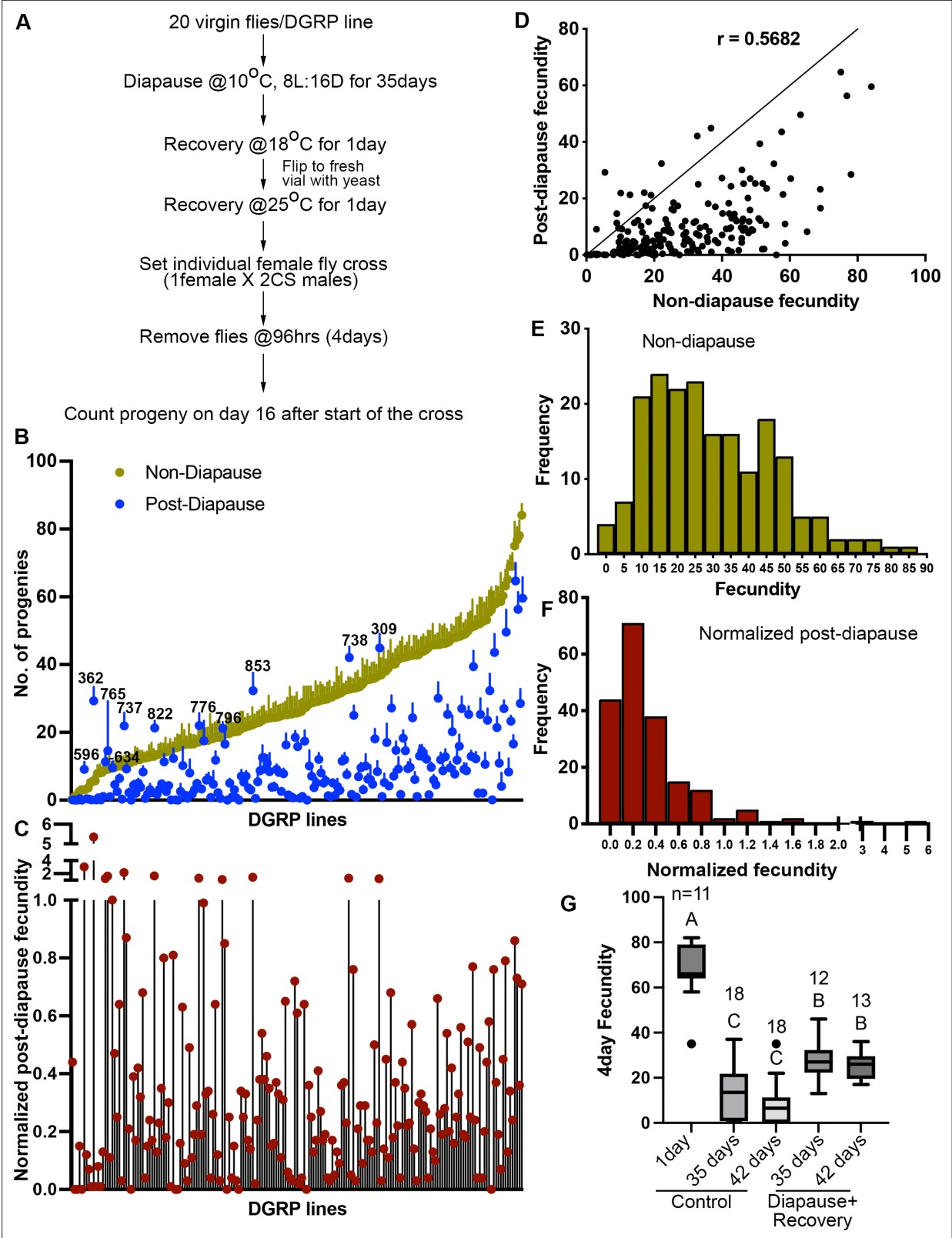

**Figure 1.** Quantification of diapause in Drosophila Genetic Reference Panel (DGRP) lines measured as the ratio of post-diapause to non-diapause fecundity. (**A**) Schematic of experimental workflow. (**B**) Average number of progenies produced (fecundity) in a 4-day individual female fly mating experiment of DGRP lines either as non-diapausing (yellow) or after a 35-day post-diapause (blue) virgin flies. Each dot represents the average fecundity, and the line above represents the standard error. DGRP line numbers are indicated wherever the post-diapause fecundity exceeds the non-diapause

*Figure 1 continued on next page*

*Figure 1 continued*

fecundity. (**C**) Normalized post-diapause fecundity average of (individual post-diapause fecundity/mean non-diapause fecundity) of each DGRP line. (**D**) Correlation of post-diapause to non-diapause fecundity. Pearson's r=0.5682. $r^2$=0.3228. (**E–F**) Frequency distribution of DGRP lines fecundity under non-diapause (**E**) and of the normalized post-diapause fecundity (**F**). (**G**) Average 4-day fecundity of single female flies, each crossed with two young Canton-S male flies, aged for 1, 35, or 42 days in non-diapause conditions or kept in diapause conditions for 35 or 42 days followed by recovery. One-way ANOVA and Tukey's multiple comparison test, compact letter display shows comparisons. n is the number of individual female fly fecundity measured, and whiskers represent the smallest and largest values within 1.5× the interquartile range (IQR).

The online version of this article includes the following source data for figure 1:

**Source data 1.** Non-diapause and post-diapause fecundity in Drosophila Genetic Reference Panel (DGRP) lines.

**Source data 2.** Excel sheet containing data corresponding to *Figure 1D–F*.

**Source data 3.** Fecundity of Canton-S control flies scored at different ages and conditions.

predicted functional relationships between genes. Additionally, it identifies subnetworks of genes related by functional Gene Ontology (GO). Enriched subnetworks are identified by dividing the number of genes from the input set by the total number of genes associated with a specific gene ontology. The subnetwork analysis is presented in *Figure 2D*, *Figure 2—source data 1*. Two primary classes of GO terms are significantly enriched in the diapause GWAS set compared to the genome as a whole: nervous system development and the reproductive system/gonad development. These two categories are consistent with the idea that changes in the environment are sensed by the nervous system, and the information is relayed to the reproductive system.

We also compared the list of diapause-associated genes to results of other *Drosophila* GWAS (*Figure 3*, *Figure 3—source data 1*). Intriguingly, the most significant overlap was with genes identified as associated with olfactory behavior (*Horváth and Kalinka, 2018*), which has not previously been described as important for diapause. The second most significant overlap was with genes identified in non-diapause fecundity and lifespan (*Durham et al., 2014*). Less significant overlaps were found with genes associated with circadian rhythm (*Harbison et al., 2019*), chill coma recovery (*Mackay et al., 2012*), development time associated with lead toxicity (*Zhou et al., 2016*), alcohol sensitivity (*Morozova et al., 2015*), starvation resistance (*Mackay et al., 2012*), mean food intake (*Garlapow et al., 2015*), sleep (*Harbison et al., 2013*), neurotoxin methylmercury tolerance (*Montgomery et al., 2014*), brain wiring regulators (*Li et al., 2020*), and death due to traumatic brain injury (*Katzenberger et al., 2015*). Overlaps with genes associated with courtship, viability in lead toxicity (*Zhou et al., 2016*), effects of genetic architecture and Wolbachia status on starvation (*Huang et al., 2014*), aggression (*Shorter et al., 2015*), ER stress (*Chow et al., 2013*), leg development (*Grubbs et al., 2013*), startle response (*Mackay et al., 2012*), and embryo development time (*Horváth et al., 2016*) were not statistically significant. In summary, both analyses implicated the nervous system, and potentially the olfactory system in particular, in the regulation of post-diapause fecundity.

## RNAi analysis of GWAS candidates identifies neural genes required for recovery

To assess the functional significance of the top candidates, we conducted an RNAi screen. First, we evaluated the effectiveness of *Gal4* under diapausing conditions. As a control, we crossed *Mat-α-tub-Gal4*, which drives expression in the female germline to *UASp-F-tractin.tdTomato* to assess the effectiveness of *Gal4*-mediated expression at 10°C. We compared tdTomato expression in flies maintained at different temperatures for 3 weeks. Compared to 25°C (*Figure 4A*) and 18°C (*Figure 4B*), flies at 10°C exhibited as high or higher expression of tdTomato (*Figure 4C*).

To evaluate the effectiveness of *Gal4*-mediated RNAi at 10°C, we used *Mat-α-tub-Gal4* to drive *zpg* (*zero population growth*, aka *INX4*) RNAi and assessed the extent of Zpg knockdown by immunostaining with an anti-Zpg antibody. We used *Mat-α-tub-Gal4* because Zpg is expressed in the germline, and we chose to target Zpg because of the availability of the anti-Zpg antibody. Zpg is expressed from the earliest stages of germline development in the germarium (*Figure 4E and E'*). *Mat-α-tub-Gal4* is not expressed detectably in the germarium but is expressed in early egg chambers (*Figure 4A–C*). So, to quantify the knockdown efficiency, we measured the level of Zpg in stage 3 egg chambers relative to the germarium staining, as an internal control. Remarkably, comparison of *Mat-α-tub-Gal4>zpg* RNAi

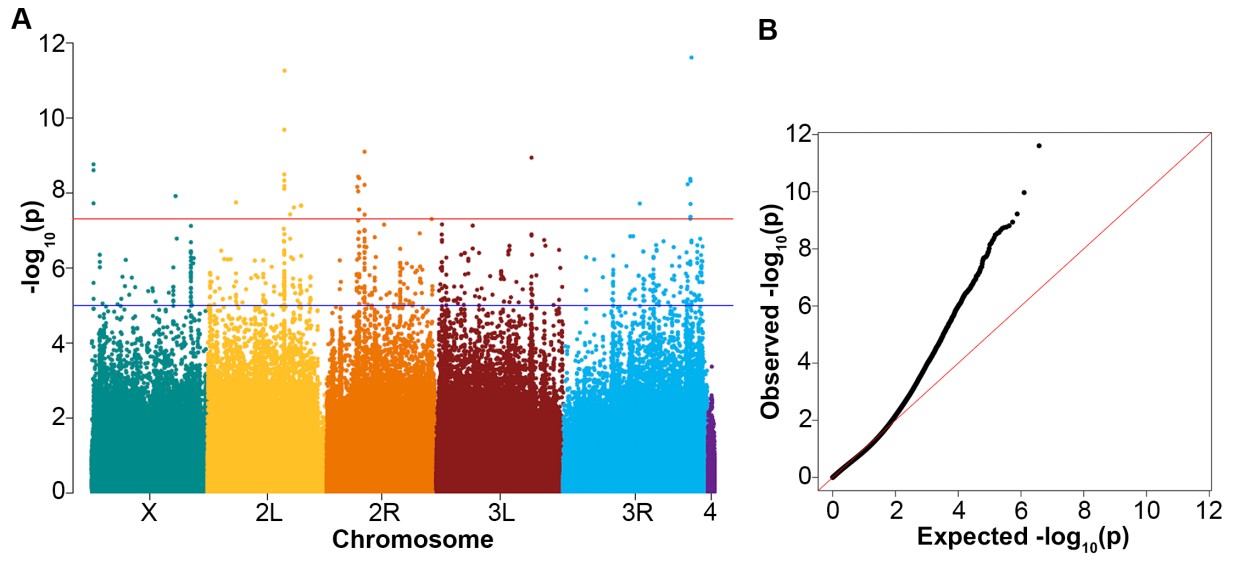

| GWAS output | Number |
| --- | --- |
| Associated genetic variants (SNPs/insertions/deletions) | 546 |
| Candidate genes (Associations in genic region +1kb up- or down-stream) | 291 |
| Genes with >1 associations | 89 |

**C**

**D**

| Description | q-value | Occurrences in sample | Occurrences in genome | Fold enrichment |
| --- | --- | --- | --- | --- |
| Axon development | 4.33E-08 | 28 | 288 | 4.6 |
| Axon guidance | 1.24E-07 | 24 | 233 | 4.9 |
| Chemotaxis | 1.24E-07 | 25 | 252 | 4.7 |
| Axonogenesis | 1.24E-07 | 27 | 290 | 4.4 |
| Neuron projection guidance | 2.01E-07 | 24 | 241 | 4.7 |
| Development of primary sexual characteristics | 6.41E-05 | 12 | 76 | 7.4 |
| Reproductive system development | 1.81E-04 | 12 | 87 | 6.5 |
| Sex differentiation | 1.81E-04 | 13 | 102 | 6.0 |
| Reproductive structure development | 1.81E-04 | 12 | 87 | 6.5 |
| Central nervous system development | 1.81E-04 | 19 | 225 | 4.0 |
| Gonad development | 2.16E-04 | 10 | 58 | 8.1 |

**Figure 2.** Genome-wide association of *Drosophila* diapause. (**A**) Manhattan plot for genome-wide association distribution. The position of each point along the y-axis indicates $-\log_{10}$(p-value) of association of a single nucleotide polymorphism (SNP), insertion, or deletion. Points above the blue line have a p-value<$1e^{-5}$. The red line represents Bonferroni-corrected p-value = $4.8e^{-8}$. (**B**) Q-Q plot of p-values from the Drosophila Genetic Reference Panel (DGRP) single variant genome-wide association study (GWAS) with the red line representing expected p-value and observed p-values deviating (black dots) from expected. (**C**) Numbers of genetic variants and candidate genes associated with diapause according to the GWAS. (**D**) Subnetworks from the Cytoscape analysis showing q-value (using the Benjamini-Hochberg procedure) for each subnetwork identified.

The online version of this article includes the following source data for figure 2:

*Figure 2 continued on next page*

*Figure 2 continued*

**Source data 1.** Gene variants associated with post-diapause fecundity identified by genome-wide association study (GWAS).

**Source data 2.** Candidate diapause-associated genes identified by genome-wide association study (GWAS).

flies kept at different temperatures for 3 weeks revealed that *Gal4*-mediated RNAi was as effective at 10°C, as it was at 18°C and 25°C (*Figure 4D–H'*; *Figure 4—source data 1*).

Fifteen GWAS hits for which multiple RNAi lines were available were selected for RNAi screening (*Figure 4—source data 2*). We used *tub-Gal4* because it is widely expressed, including in neurons. We measured non-diapause and post-diapause fecundity. For the five RNAi lines that proved lethal, we used *tubulin-Gal80^ts* to suppress the RNAi expression during development and diapause and then shifted the flies to 30°C to inactivate the Gal80 and drive the RNAi during recovery (*Figure 4I*), and measured post-diapause fecundity. In this way, we could test specifically for a function in post-diapause recovery (*Figure 4—source data 2*). RNAi against two genes, *Defective-proboscis-extension-response interacting protein-γ* (Dip-γ) and *Scribbler* (*sbb*), significantly reduced post-diapause fecundity (*Figure 4J*).

To determine whether Dip-γ and *sbb* are required specifically in neurons, we crossed *nSyb-Gal4*, a pan-neuronal driver, to validated *UAS-Dip-γ* RNAi and *UAS-sbb* RNAi lines (*Davis et al., 2014*; *Shimozono et al., 2019*). Neuron-specific knockdown of Dip-γ (*Figure 4K*, *Figure 4—source data 3*) caused as severe a defect in post-diapause fecundity as ubiquitous RNAi, in contrast to glial knockdown using *Repo-Gal4* (*Figure 4L*, *Figure 4—source data 3*). Pan-neuronal RNAi of *sbb* (*Figure 4K*) also significantly inhibited post-diapause fecundity more than glial RNAi (*Figure 4L*). We conclude that Dip-γ and *sbb* are required in neurons for successful post-diapause fecundity, consistent with the enrichment of this gene class in the diapause GWAS.

## Post-diapause fecundity requires neurons in the antenna

While we were able to show significant functional effects of Dip-γ and *sbb*, GWASs by their nature identify many genes with small effects, which are too small to detect individually. Furthermore, the overlap with genes associated with olfactory behavior led us to complement the genetic analysis by testing whether the antenna and neurons within it are required for successful diapause. We removed the antenna from CS flies and measured post-diapause fecundity. As a control for the surgery, we removed the arista, which is an appendage from the antenna. Removal of the antenna but not the arista reduced post-diapause fecundity compared to unmanipulated controls (*Figure 5A*, *Figure 5—source data 1*). Removal of the antenna but not the arista also reduced the number of germline stem cells (GSCs) post-diapause (*Figure 5B*). These results suggest that sensory cells in the antenna are important for successful recovery post-diapause.

Another important diapause trait is lifespan extension. So, we tested the effect of removing the antenna on post-diapause and non-diapause lifespan (*Figure 5C and D*). Antenna removal had little effect on non-diapause lifespan (median survival 72 days w/o antenna vs 75 days with antenna) (*Figure 5C*). In contrast, antennaless flies exhibited a substantial reduction in diapause lifespan compared to flies with antennae (*Figure 5C*). We considered the possibility that the precipitous deaths of many antennaless flies at 10°C (*Figure 5C*) might be a consequence of impaired wound healing. So, we repeated the experiments, allowing the flies 2 weeks to recover at 25°C prior to moving them to the diapause-inducing temperature of 10°C (*Figure 5D*). Two weeks at 25°C substantially prevented the early post-surgery death. Yet, antennaless flies still had shorter lifespans (median survival 95 days) compared to intact flies (median survival 142 days), supporting the importance of the antenna in lifespan extension (see Discussion). A potential caveat for the latter experiment is that 2 weeks at 25°C might impair diapause induction. However, flies with antennae exhibited very similar lifespan extension at 10°C with or without the 2-week wound-healing period at 25°C (compare dark green lines in *Figure 5C and D*), mitigating that concern. We conclude that the antennae are likely important for lifespan extension and post-diapause fecundity.

*Drosophila* antennae are involved in various sensory modalities, including olfaction. To identify which neurons are required for diapause, we suppressed neuronal activity in subsets of cells by expressing the tetanus toxin light chain protein (TNT), which selectively cleaves the neuronal isoform of fly synaptobrevin. Orco is the virtually universal co-receptor for odorant receptors of the OR class (*Montell, 2021*), so we used Orco-*Gal4* to drive expression. We found that blocking neuronal transmission in the Orco-expressing neurons decreased the post-diapause/non-diapause fecundity ratio

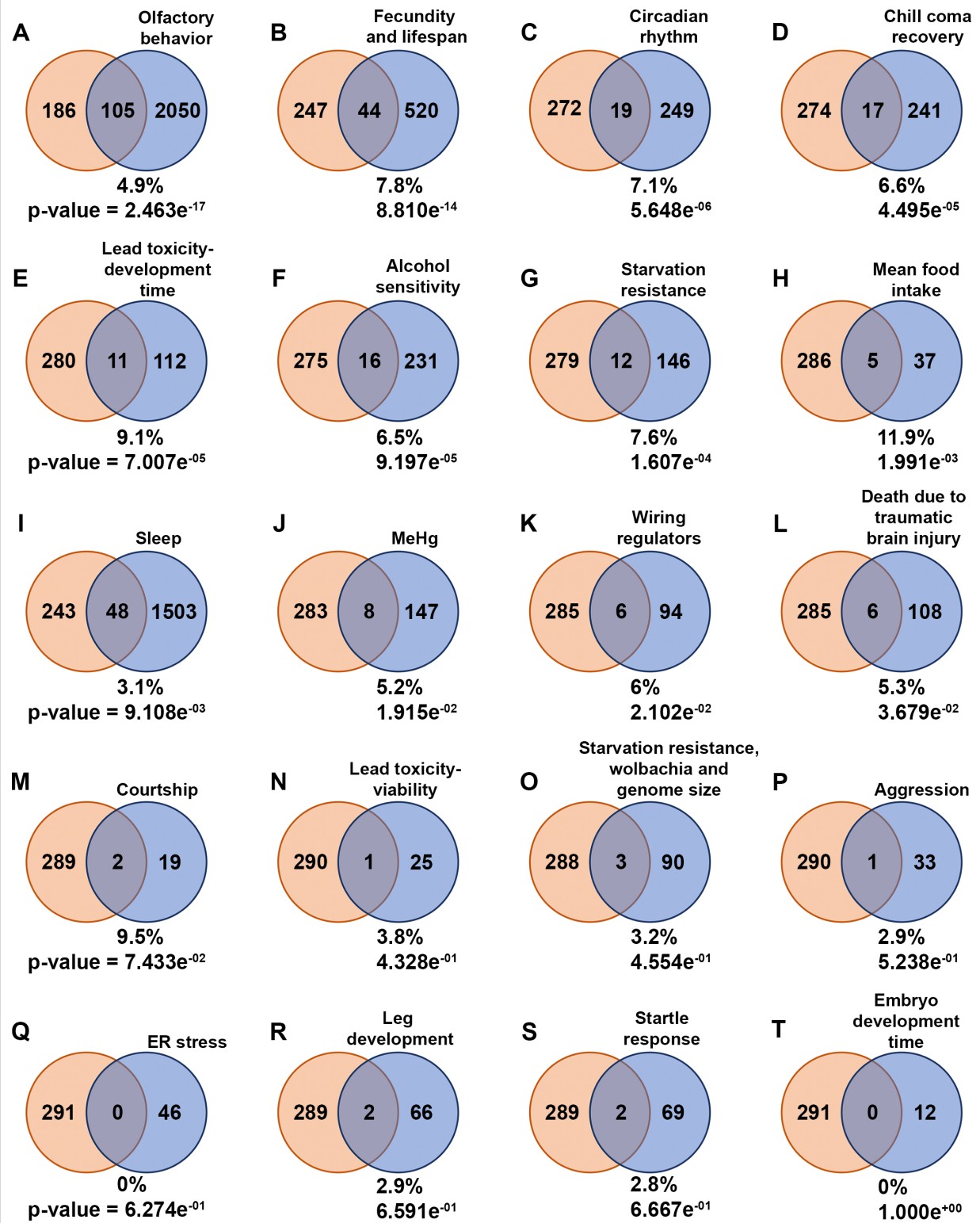

**Figure 3.** Common genes to diapause-genome-wide association study (GWAS) hits and other behavior-associated genes. (**A–T**) Venn diagrams illustrate the intersection of genes associated with diapause identified through GWAS (diapause-GWAS) with genes from other behavior-related gene lists obtained from various studies. The diapause-GWAS gene set is represented as the first set throughout the figure, while subsequent sets represent different behavior-related gene lists identified in separate studies. The percentage of common genes compared to the total genes from different

*Figure 3 continued on next page*

*Figure 3 continued*

respective behavior-associated gene lists is provided for each Venn diagram. p-Values of overlap to the diapause gene list determined by Fisher's exact tests are also provided. Venn diagrams are arranged in the order of p-values.

The online version of this article includes the following source data for figure 3:

**Source data 1.** Comparison of diapause-genome-wide association study (GWAS) gene list to other behavior-associated gene lists.

(*Figure 5E*, *Figure 5—source data 3*), further implicating odor perception in successful diapause recovery (see Discussion).

Ionotropic receptors (IRs) are another class of odorant receptor (*Montell, 2021*). Ir25a is broadly expressed, and inhibiting Ir25a-expressing neurons by expressing TNT reduced the post-diapause/non-diapause fecundity ratio compared to *Gal4*s expressed in other IR-expressing cells such as Ir8a and Ir84a, which are involved in odor perception, and Ir76b, which is involved in contact chemosensation (*Figure 5F*, *Figure 5—source data 3*). We also found that inhibiting neurons expressing Ir21a, mutations in which disrupt warm and cool avoidance behaviors (*Budelli et al., 2019*), dramatically impaired recovery from diapause. Blocking Ir21a neuronal transmission prevented flies from exiting diapause, as evidenced by 100% mortality of the post-diapause female flies moved to recovery conditions (n=25). Expressing TNT in the 'hot cell' neurons (*Ni et al., 2013*), which respond to heating, using *HC-Gal4*, also significantly reduced the post-diapause fecundity (*Figure 5F*, *Figure 5—source data 3*).

## Discussion

### Genome-wide analysis of the effect of diapause on fecundity

*Drosophila* diapause is a fascinating life history trait that confers resilience in stressful environments and extends reproductive potential and organismal longevity. Our GWAS identified nearly 300 candidate genes associated with an important but understudied feature of diapause: the ability to recover and reproduce successfully post-diapause. The most striking finding is that genes associated with neural development are highly overrepresented in the diapause set. We further confirmed that at least two such genes, Dip-γ and *sbb*, are essential for post-diapause fecundity.

When we started this project, little was known about the neural control of diapause. Consistent with our finding reported here that neural genes are enriched in the diapause GWAS, and our recent finding that circadian activity and sleep are dramatically altered in diapause (*Meyerhof et al., 2024*), two recent studies have reported that low temperature affects two subsets of circadian neurons, DN3s, and sLNvs, which in turn impact reproductive arrest (*Hidalgo et al., 2023*; *Meiselman et al., 2022*). sLNv neurons secrete the neuropeptide pigment dispersing factor (PDF) onto insulin-producing cells (IPCs), which secrete insulin-like peptides, which in turn promote JH production. At 10°C, PDF mRNA and protein levels are reduced in sLNv neurons, reducing JH production, which arrests vitellogenesis. In parallel, cool-temperature-induced reduction in DN3 activity disinhibits cholinergic ASTC-R2 neurons, which in turn inhibit vitellogenic egg chamber development. These neurons do not synapse onto the IPCs, and the mechanism by which the ASTC-R2 neurons affect oogenesis is unknown. Thus, there are at least two parallel mechanisms by which cool temperatures affect neural activity and contribute to reproductive arrest. It is not yet clear how cool temperatures reduce PDF mRNA or how PDF mRNA and protein levels rebound during recovery.

We found that the neuronal transcription factor Sbb is required post-diapause for recovery, so it is interesting to speculate that Sbb could affect PDF mRNA abundance. Sbb functions predominantly as a transcriptional repressor, so it could promote PDF expression in recovery by inhibiting a repressor of PDF. Alternatively, PDF reduces levels of the transcriptional co-activator Eyes Absent (Eya) in the IPCs, so reduced PDF leads to elevated Eya protein levels during diapause. It is possible that Sbb represses Eya transcription to facilitate recovery post-diapause. A third possibility is that Sbb functions in the DN3-ASTC-R2 pathway. We have also found that circadian rhythms and sleep behavior are dramatically altered in flies at 10°C (*Meyerhof et al., 2024*). It will be of interest to determine if Dip-γ and *sbb* are required for those behavioral effects at low temperatures, in addition to the effects on post-diapause fecundity described here.

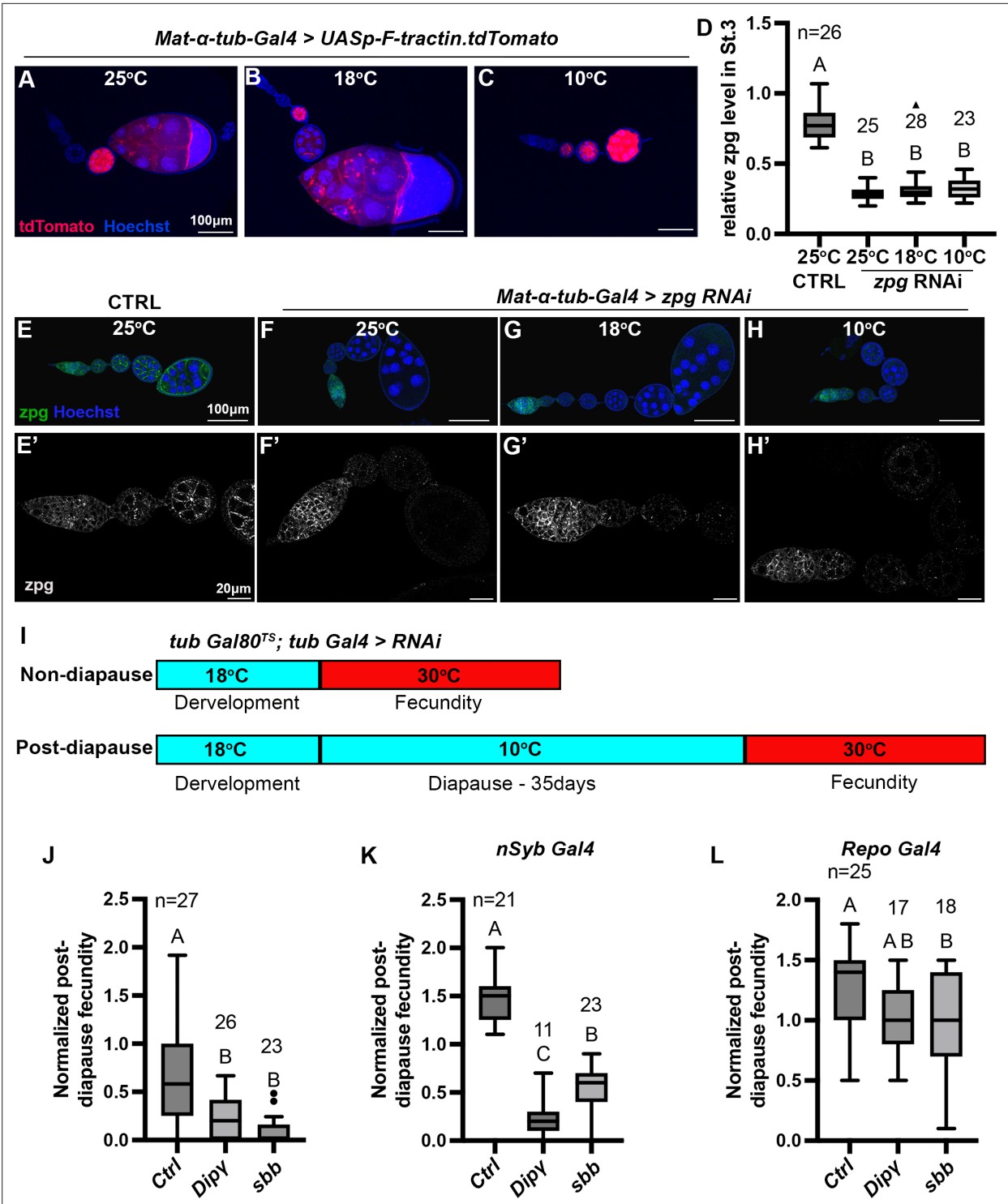

**Figure 4.** RNAi-mediated loss-of-function study to identify genes involved in diapause. (**A–C**) *Mat-α-tub-Gal4* driving expression of *UASp-F-tractin.tdTomato* (red) at the indicated temperatures. Scale bars are 100 μm. (**D**) Quantification of *zpg* RNAi knockdown in stage 3 egg chambers normalized to the level of Zpg in the germarium (one-way ANOVA and Tukey's multiple comparison test, compact letter display shows comparisons). The numbers (**n**) of stage 3 egg chambers quantified are shown, and whiskers represent the smallest and largest values within 1.5× the interquartile range (IQR). (**E–H**) Representative images of egg chambers stained with anti-Zpg antibody (green) from either control (no-knockdown) (**E, E'**) or knockdown of *zpg* (**F-H'**) driven by *Mat-α-tub-Gal4* at different temperatures. (**E'–H'**) are higher magnification, single channel views of the ovarioles shown in (**E–H**). Scale bars are 100 μm in (**E–H**) and 20 μm in (**E'–H'**). All flies in (**A–H'**) were kept at respective temperatures for 3 weeks. (**I**) Experimental design for RNAi knockdown specifically during recovery for the experiment shown in (**J**). The temperature-sensitive Gal80 repressor of Gal4 prevented RNAi expression

*Figure 4 continued on next page*

*Figure 4 continued*

during development and diapause. Incubation at 30°C during recovery inactivates Gal80, allowing Gal4-mediated RNAi knockdown. (**J**) Ubiquitous knockdown of *Dip-γ* or *sbb* with *tub Gal4* specifically during recovery as shown in (**I**) significantly reduces post-diapause/non-diapause fecundity compared to the control (*tubGal80^{TS};tubGal4 >Ctrl* RNAi #9331). (**K**) Pan-neuronal RNAi knockdown of *Dip-γ* and *sbb* with *nSybGal4* significantly reduces post-diapause/non-diapause fecundity compared to the control (*nSyb Gal4>Ctrl* RNAi #54037). (**L**) Glia-specific knockdown of *Dip-γ* or *sbb* with *Repo Gal4* causes little or no reduction in post-diapause/non-diapause fecundity (Control- *Repo Gal4>Ctrl* RNAi #54037). In (**J–L**), one-way ANOVA and Tukey's multiple comparison test, compact letter display shows comparisons. n is the number of individual female flies tested, and whiskers represent the smallest and largest values within 1.5× the interquartile range (IQR).

The online version of this article includes the following source data for figure 4:

**Source data 1.** Excel sheet containing data corresponding to *Figure 4D*.

**Source data 2.** RNAi screen of top 15 genome-wide association study (GWAS)-associated genes.

**Source data 3.** Tissue-specific RNAi of Dip-γ and *sbb*.

## Olfactory neurons are required for post-diapause fecundity

We complemented our genetic analysis with an investigation of the cells that contribute to diapause. The overlap between olfactory-behavior-associated genes and diapause-associated genes inspired us to test if the antenna is required, and we found that the antenna is required both for post-diapause fecundity and for lifespan extension. We conclude that cells in the antenna transmit important information for successful diapause. Furthermore, we found that inhibiting olfactory receptor neurons by expressing TNT with Orco-Gal4 caused a similar effect on post-diapause fecundity as antenna removal. This is interesting because Meiselman et al. found that the antenna was not required for accumulation of mature eggs post-diapause (*Meiselman et al., 2022*). Our measurement of fecundity required not only that mature eggs developed in the ovary but also that they were fertilized, laid, and could develop into adults. So, what is required for production of viable progeny in addition to mature eggs? Since both feeding and mating are suppressed during diapause, the results suggest that the antenna may be required to promote these behaviors, consistent with the known role of the antenna in providing sensory information important for flies to seek food and mates (*Montell, 2021*; *Tunstall et al., 2012*). The antenna may be dispensable for relieving the vitellogenesis block in the ovary but may specifically be required for animals to find sufficient food and mates post-diapause to support maximal fecundity.

Consistent with the possible role of the olfactory system in diapause traits, in a screen of 505 gene trap lines, Tunstall et al. identified 16 genes that were expressed in ORNs, three of which are present in our diapause-associated gene set: *tai, cpo,* and sbb (*Tunstall et al., 2012*). Cpo is a known diapause-associated gene (*Kankare et al., 2012*), and we show here that sbb is also required for post-diapause fecundity. Tunstall et al. showed that *tai* is required for normal olfactory-mediated attraction to food. The olfactory system also regulates longevity in *Caenorhabditis elegans* (*Alcedo and Kenyon, 2004*), as well as *Drosophila* (*Libert et al., 2007*), and the precise effects depend on environmental factors, including temperature and availability of food and mates (*Allen et al., 2015*), consistent with our findings on the role of olfaction in post-diapause lifespan extension and fecundity.

We further explored whether IR-expressing ORNs might also be required post-diapause by inhibiting subsets of IR-expressing cells with TNT. The most remarkable finding was that expressing TNT with *Ir21a-Gal4* resulted in post-diapause death, indicating that *Ir21a-Gal4*-expressing cells are essential for successful exit from dormancy. Ir21a is an IR expressed in cells sensitive to cooling (cool cells). Animals mutant for Ir21a fail to detect cooling and also fail to avoid both warm and cool temperatures (*Budelli et al., 2019*). That cells required for temperature sensation are essential for diapause is perhaps not surprising; however, it is at first glance puzzling that inhibiting the Ir21a-expressing cells affects diapause because Ir21a-expressing cells are located in the arista, which when amputated did not affect post-diapause fecundity in our experiments. Similarly puzzling is the observation that inhibition of the 'hot cells', which are also located in the arista in close proximity to the cool cells, also impairs post-diapause fecundity. One possible explanation for these results could be that cool cells and hot cells inhibit one another such that eliminating cool cell activity actually disinhibits hot cells and vice versa. In fact, *Budelli et al., 2019*, found hot cell spiking in Ir21a mutants, as well as defects in avoidance of both warm and cool temperatures. If inhibiting cooling-sensitive cells hyperactivates

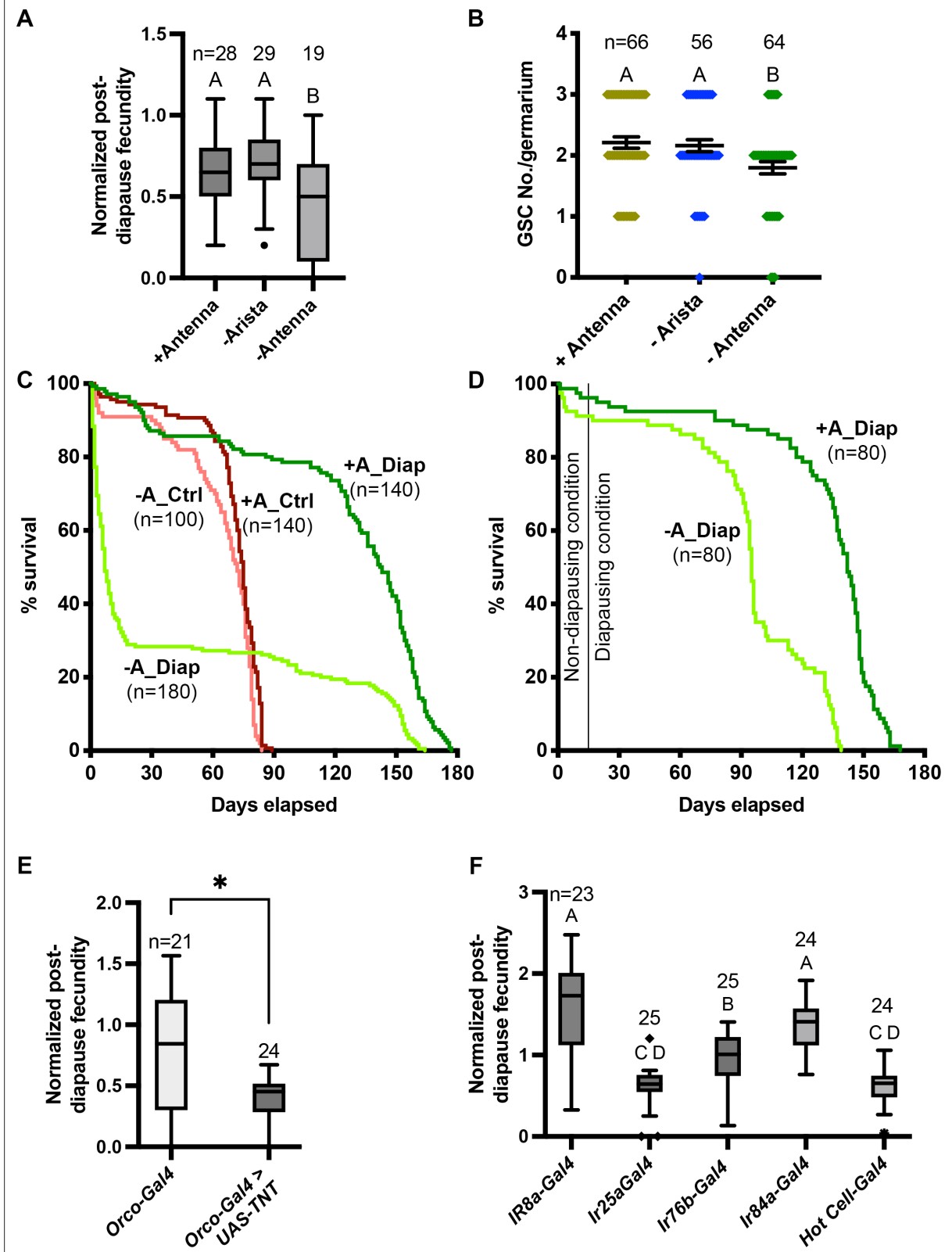

**Figure 5.** Neural control of diapause. (**A**) Effect of antenna removal on recovery of fecundity post-diapause. Arista removal was used as a control for the surgery. n is the number of individual female flies tested. One-way ANOVA and Tukey's multiple comparison test, compact letter display shows comparisons. Whiskers represent the smallest and largest values within 1.5× the interquartile range (IQR). (**B**) Effect of antenna removal on germline stem cell (GSC) recovery after 5 weeks of diapause. n is the number of germaria counted (there are typically 2–3 GSCs/germarium). One-way ANOVA

*Figure 5 continued on next page*

*Figure 5 continued*

and Tukey's multiple comparison test with compact letter display to show comparisons. Error bars represent standard error. (**C**) Role of antenna in lifespan extension in diapause. Control flies were maintained at 25°C and diapause flies were moved to 10°C. Median survival for flies with intact antenna in diapause (+A_Diap) – 142.5 days; antennaless flies in diapause (-A_Diap) – 7 days; control with intact antenna in optimal conditions (+A_Ctrl) – 75 days; and antennaless flies in optimal conditions (-A_Ctrl) – 72 days. Survival curves are compared pairwise using the Log-rank (Mantel-Cox) test and $X^2$ values are: Diap±antenna = 98.89, Ctrl ±antenna = 11.69, Diap+antenna vs Ctrl+antenna = 163.5, and Diap – antenna vs Ctrl – antenna=0.5. n represents the number of flies used for the survival curve. (**D**) Control (+A_Diap) and antennaless (-A_Diap) flies were maintained at 25°C for 2 weeks post-surgery to allow for wound healing and shifted to the diapausing conditions. Median survival for +A_Diap – 142 days; -A_Diap – 95 days. Survival curves are compared pairwise using the Log-rank (Mantel-Cox) test and $X^2$ values = 86. n represents the number of flies used for the survival curve. (**E–F**) Effects on diapause of inactivating neuronal transmission by driving tetanus toxin using UAS-TeTxLC.tdt, (**E**) in odor responsive neurons using the indicated *Gal4* lines. Orco is a co-receptor for odorant receptors of the OR class. (**F**) Ir8a is a co-receptor involved in organic acid detection. Ir25a is a co-receptor involved in chemo- and thermo-sensation. Ir76b is involved in the detection of various amines and salt. Ir84a is involved in the detection of phenylacetic acid and male courtship behavior. Hot cells are heat-sensitive cells in the arista. In (E-F), one-way ANOVA and Tukey's multiple comparison test, compact letter display shows comparisons. n is the number of individual female flies tested, and whiskers represent the smallest and largest values within 1.5× the interquartile range (IQR).

The online version of this article includes the following source data for figure 5:

**Source data 1.** Role of antennae in the ability of flies to undergo diapause.

**Source data 2.** Excel sheet containing data corresponding to *Figure 5B–D*.

**Source data 3.** Identification of specific cell types involved in diapause in fly antennae.

warming-sensitive cells and vice versa, then eliminating both cell types by removal of the arista might be less harmful than inactivating individual cell types.

In summary, we report the first GWAS of diapause, using the stringent assay of the ability to enter diapause, survive, exit diapause, and produce fertile offspring. This analysis has revealed a key role for the olfactory system, as well as specific genes and cells required for diapause lifespan extension and post-diapause fecundity. Additionally, the 291 candidate diapause-associated genes identified here provide many intriguing candidates for future study.

# Materials and methods

## Key resources table

| Reagent type (species) or resource | Designation | Source or reference | Identifiers | Additional information |
|---|---|---|---|---|
| Genetic reagent (*D. melanogaster*) | DGRP lines | BDSC; *Huang et al., 2014* | RRID:SCR_006457 | |
| Genetic reagent (*D. melanogaster*) | 2704 | BDSC | RRID:BDSC_27049 | |
| Genetic reagent (*D. melanogaster*) | 32034 | BDSC | RRID:BDSC_32034 | |
| Genetic reagent (*D. melanogaster*) | 40889 | BDSC | RRID:BDSC_40889 | |
| Genetic reagent (*D. melanogaster*) | 27508 | BDSC | RRID:BDSC_27508 | |
| Genetic reagent (*D. melanogaster*) | 54811 | BDSC | RRID:BDSC_54811 | |
| Genetic reagent (*D. melanogaster*) | 44661 | BDSC | RRID:BDSC_44661 | |
| Genetic reagent (*D. melanogaster*) | 53315 | BDSC | RRID:BDSC_53315 | |
| Genetic reagent (*D. melanogaster*) | 54037 | BDSC | RRID:BDSC_54037 | |
| Genetic reagent (*D. melanogaster*) | 57840 | BDSC | RRID:BDSC_57840 | |

*Continued on next page*

*Continued*

| Reagent type (species) or resource | Designation | Source or reference | Identifiers | Additional information |
|---|---|---|---|---|
| Genetic reagent (*D. melanogaster*) | 63013 | BDSC | RRID:BDSC_63013 | |
| Genetic reagent (*D. melanogaster*) | 56867 | BDSC | RRID:BDSC_56867 | |
| Genetic reagent (*D. melanogaster*) | 80461 | BDSC | RRID:BDSC_80461 | |
| Genetic reagent (*D. melanogaster*) | 35785 | BDSC | RRID:BDSC_35785 | |
| Genetic reagent (*D. melanogaster*) | 9331 | BDSC | RRID:BDSC_9331 | |
| Genetic reagent (*D. melanogaster*) | 5138 | BDSC | RRID:BDSC_5138 | |
| Genetic reagent (*D. melanogaster*) | 7019 | BDSC | RRID:BDSC_7019 | |
| Genetic reagent (*D. melanogaster*) | 23292 | BDSC | RRID:BDSC_23292 | |
| Genetic reagent (*D. melanogaster*) | 41731 | BDSC | RRID:BDSC_41731 | |
| Genetic reagent (*D. melanogaster*) | 41728 | BDSC | RRID:BDSC_41728 | |
| Genetic reagent (*D. melanogaster*) | 51311 | BDSC | RRID:BDSC_51311 | |
| Genetic reagent (*D. melanogaster*) | 41750 | BDSC | RRID:BDSC_41750 | |
| Genetic reagent (*D. melanogaster*) | 28838 | BDSC | RRID:BDSC_28838 | |
| Genetic reagent (*D. melanogaster*) | 35607 | BDSC | RRID:BDSC_35607 | |
| Genetic reagent (*D. melanogaster*) | 7063 | BDSC | RRID:BDSC_7063 | |
| Genetic reagent (*D. melanogaster*) | 58989 | BDSC | RRID:BDSC_58989 | |
| Genetic reagent (*D. melanogaster*) | 51635 | BDSC | RRID:BDSC_51635 | |
| Genetic reagent (*D. melanogaster*) | 64349 | BDSC | RRID:BDSC_64349 | |
| Genetic reagent (*D. melanogaster*) | 7415 | BDSC | RRID:BDSC_7415 | |
| Genetic reagent (*D. melanogaster*) | 44077 | VDRC | RRID:Flybase_FBst046539 | |
| Genetic reagent (*D. melanogaster*) | 330386 | VDRC | RRID:Flybase_FBst0491275 | |
| Genetic reagent (*D. melanogaster*) | 10268 | VDRC | RRID:Flybase_FBst0450061 | |
| Genetic reagent (*D. melanogaster*) | 40698 | VDRC | RRID:Flybase_FBst0463709 | |
| Genetic reagent (*D. melanogaster*) | 991 | VDRC | RRID:Flybase_FBst0471623 | |

*Continued on next page*

*Continued*

| Reagent type (species) or resource | Designation | Source or reference | Identifiers | Additional information |
|---|---|---|---|---|
| Genetic reagent (*D. melanogaster*) | 992 | VDRC | RRID:Flybase_FBst0471628 | |
| Genetic reagent (*D. melanogaster*) | 21052 | VDRC | RRID:Flybase_FBst0453901 | |
| Genetic reagent (*D. melanogaster*) | 100412 | VDRC | RRID:Flybase_FBst0472285 | |
| Genetic reagent (*D. melanogaster*) | 51936 | VDRC | RRID:Flybase_FBst0469629 | |
| Genetic reagent (*D. melanogaster*) | 106911 | VDRC | RRID:Flybase_FBst0478734 | |
| Genetic reagent (*D. melanogaster*) | 104551 | VDRC | RRID:Flybase_FBst0478734 | |
| Genetic reagent (*D. melanogaster*) | 17767 | VDRC | RRID:Flybase_FBst0476409 | |
| Genetic reagent (*D. melanogaster*) | 104056 | VDRC | RRID:Flybase_FBst0452858 | |
| Genetic reagent (*D. melanogaster*) | 60102 | VDRC | RRID:Flybase_FBst0475914 | |
| Genetic reagent (*D. melanogaster*) | 013805 | VDRC | RRID:SCR_013805 | |
| Genetic reagent (*D. melanogaster*) | 101659 | VDRC | RRID:Flybase_FBst0473532 | |
| Genetic reagent (*D. melanogaster*) | 36166 | VDRC | RRID:Flybase_FBst0461553 | |
| Genetic reagent (*D. melanogaster*) | *Hot cell-Gal4* | Zuker lab, *Gallio et al., 2011* | DOI: 10.1016 /j.cell.2011.01.028 | |
| Genetic reagent (*D. melanogaster*) | *Ir21a-Gal4* | Craig Montell lab, *Ni et al., 2016* | DOI: https://doi.org/10.7554/eLife.13254 | |
| Antibody | anti-Zpg antibody (rabbit polyclonal) | *Smendziuk et al., 2015* | DOI: https://doi.org/10.1242/dev.123448 | 1:20,000 |
| Antibody | anti-Hts antibody (1B1) (mouse monoclonal) | DSHB | RRID:AB_528070 | 1:20 |
| Antibody | anti-Vasa antibody (rat monoclonal) | DSHB | RRID:AB_760351 | 1:20 |
| Software, algorithm | Prism | GraphPad Version 10.4.2 (534) | RRID:SCR_002798 | |
| Software, algorithm | Cytoscape | *Shannon et al., 2003* | RRID:SCR_003032 | |
| Software, algorithm | GeneMANIA | *Warde-Farley et al., 2010* | RRID:SCR_005709 | |
| Software, algorithm | Set Comparison Appyter | *Clarke et al., 2021* | RRID:SCR_021245 | |
| Software, algorithm | DGRP2 | *Mackay et al., 2012* | http://dgrp2.gnets.ncsu.edu/ | |

### *Drosophila* stocks

The majority of the experiments were carried out using the inbred, sequenced lines of the DGRP (*Huang et al., 2014*). Diapause trait quantified in 193 DGRP lines was used in the GWAS analysis.

Please refer to *Figure 1—source data 1*. The RNAi lines and *Gal4* stocks were from BDSC and VDRC (v44077, v330386, 27049, v10268, v40698, 32034, v101659, v36166, 40889, 27508, v991, v992, v21052, v100412, 54811, v51936, v106911, 44661, 53315, 54037, 57840, v104551, v17767, 63013, 56867, 80461, v104056, 35785, v60102, 9331, 5138, 7019, 23292, 41731, 41728, 51311, 41750, 28838, 35607, 7063, 58989, 51635, 64349, and 7415). *Hot cell-Gal4* (*Gallio et al., 2011*) is a gift from Zuker lab, and *Ir21a-Gal4* (*Ni et al., 2016*) is a gift from Craig Montell lab.

## Fecundity assay

For the non-diapause fecundity assessment, a minimum of 15 newly eclosed, not more than 6-hr-old virgin females of each strain were collected under mild $CO_2$ anesthesia. These females were individually placed into fly food vials containing cornmeal, molasses, agar medium, and yeast. In individual female fly crosses, one female from the DGRP line and two male flies of the same genotype (CS) were added to each vial.

In the post-diapause group, a minimum of 20 newly eclosed, not more than 6-hr-old virgin female flies of each strain were collected and promptly transferred to a cold room under diapause conditions (10°C and 8L:16D) for 5 weeks. After this diapause induction period, the flies were shifted to 18°C and 12L:12D for 1 day for temperature acclimation. Subsequently, the flies were transferred to new vials with fresh food medium dusted with yeast and placed at 25°C and 12L:12D for an additional day. Similar to the non-diapause crosses, individual female fly crosses were set up. The flies were allowed to mate and lay eggs for 4 days at 25°C under a normal photoperiod of 12L:12D. After 4 days, the parental flies were discarded, and the resulting progeny were counted on the 16th day from the initial mating start date. The progeny count was limited to the number of adult flies and any black pupae produced by a single female during these 16 days. If female flies failed to survive diapause, the line was given the lowest score of zero, indicating an inability to diapause. Those who died during the 4-day cross-period were not considered for scoring. In cases of low replicates due to deaths during diapause, recovery, or cross, the process was repeated to obtain sufficient replicates. The fecundity of flies can vary with the age of fly food, dryness of fly food, age of male flies used to assess the female fecundity, crowding in the bottle, etc. To control all these variables, we have always used fresh fly food (1-week-old) and 1-week-old male flies to reduce the variation. Also, we control the number of flies in collection bottles to 30–40 – about 25–30 females and 10 males and flipped every 3–4 days to a new bottle. We have also observed the fly vials regularly to prevent dryness of food and added water whenever necessary.

Similarly, assessments of non-diapause and post-diapause fecundity were conducted for the RNAi experiments. The post-diapause fecundity was normalized by dividing individual female fly fecundity by average non-diapause fecundity of the same genotype.

## Scoring of fecundity

For both the non-diapause and the post-diapause, the fecundity of each strain was calculated based on the average number of progeny produced across all replicates, excluding cases in which the female and/or both males died during the 4-day mating window. To accurately assess the diapause capacity of each strain, we normalized it by dividing the individual post-diapause fecundity by the average of non-diapause fecundity for each DGRP line. The average of this normalized post-diapause fecundity was then utilized in our GWAS analysis pipeline (*Mackay et al., 2012*). This normalization eliminated the basal difference in fecundity among DGRP lines, allowing us to focus on the variability in diapause. We have also performed a one-way ANOVA to get the mean squares for between-group and within-group variances and calculated broad-sense heritability using the formula: $H^2 = MS_G \ MSE/MS_G + (k–1) MS_E$, where $MS_G$ – mean square between groups and $MS_E$ – mean square within groups and k – number of individuals per group. Using this formula, the broad-sense heritability for normalized post-diapause fecundity was found to be 0.51.

## Data analysis

GWAS was carried out using the DGRP2 website. We conducted GO enrichment and network analyses based on the top variants ($p < 1e10^{-5}$) associated with the mean post-diapause fecundity/non-diapause fecundity score using the GeneMANIA application in Cytoscape (*Shannon et al., 2003*; *Warde-Farley et al., 2010*). The GO categories and q-values from the false discovery

rate-corrected hypergeometric test for enrichment are reported. Additionally, coverage ratios for the number of annotated genes in the displayed network vs the number of genes with that annotation in the genome are provided. GeneMANIA estimates q-values using the Benjamini-Hochberg procedure.

Set Comparison Appyter was used for comparing the diapause-GWAS gene list to other behavior gene lists using 13,500 as the background gene list and 0.05 as the significance level to calculate the p-value (*Clarke et al., 2021*).

## Immunostaining

Immunofluorescence was performed using standard procedures. Briefly, adult female ovarioles were carefully dissected in phosphate-buffered saline (PBS) using a bent tungsten needle, pulling on the stalk region of older egg chambers to minimize damage to the germarium. Ovarioles prepared for immunostaining were fixed in 4% paraformaldehyde in PBS for 20 min. To prevent sample sticking and facilitate settling, 20 µl of PBST (PBS+0.2% Triton X-100) was added to the fixing solution. After fixation, ovarioles were washed three times for 10 min each in PBST and then incubated with primary antibodies at 4°C overnight. The following morning, ovarioles were washed twice for 15 min in PBST before incubation with secondary antibodies for at least 2 hr at room temperature. After the removal of secondary antibodies, the samples were washed three times in PBST for 10 min each. Hoechst was used as a nuclear stain and added to the second wash solution. The ovarioles were subsequently mounted in Vectashield and stored at 4°C until imaging. All antibody dilutions were prepared in PBST. Due to difficulties in settling diapause samples in the solutions, a 5 min hold step was introduced on a stand before removing solutions from the tube at each step, consistently followed for all controls as well. The anti-Zpg antibody (rabbit polyclonal) was used at a 1:20,000 dilution (*Smendziuk et al., 2015*). The antibodies used for identifying GSCs are Hts (1B1) and Vasa (DSHB) (*Easwaran et al., 2022*).

## Lifespan analysis

Virgin female flies were collected and moved to the conditions to assess their lifespan. In the antenna-removed fly lifespan measurement, before moving to the diapausing conditions, we gave 14 days at optimal conditions (25°C, 12L:12D photoperiod) to heal the wound caused by amputation of the antenna. Flies were kept in food vials with a density of 20 or fewer flies per vial. Deaths were censored daily, and all vials were flipped every other day for lifespan measurement at optimal condition (25°C, 12L:12D photoperiod) and once every month for lifespan measurement at diapause condition (10°C, 16L:8D photoperiod) to prevent desiccation and drying up of fly food. The survival plot was created in Prism using the Kaplan-Meier survival analysis. Survival curves are compared pairwise by the Log-rank (Mantel-Cox) test to obtain the $\chi^2$-values.

## Acknowledgements

We thank Dominique Houston, Mackenzie Kui, Yishi Xu, and Alyssa Chow for technical assistance and members of the lab for discussions. We thank Maddalina Nano for proofreading the manuscript. This work was supported by NIH grant R01AG36907 to DJM.

## Additional information

### Funding

| Funder | Grant reference number | Author |
| --- | --- | --- |
| National Institutes of Health | R01AG36907 | Denise J Montell |

The funders had no role in study design, data collection and interpretation, or the decision to submit the work for publication.

## Author contributions
Sreesankar Easwaran, Conceptualization, Data curation, Formal analysis, Investigation, Visualization, Methodology, Writing – review and editing; Denise J Montell, Conceptualization, Formal analysis, Supervision, Funding acquisition, Writing - original draft, Project administration, Writing – review and editing

## Author ORCIDs
Sreesankar Easwaran (iD) https://orcid.org/0000-0003-4676-2150
Denise J Montell (iD) https://orcid.org/0000-0001-8924-5925

Reviewer #1 (Public review): https://doi.org/10.7554/eLife.98142.4.sa1
Author response https://doi.org/10.7554/eLife.98142.4.sa2

# Additional files

## Supplementary files
MDAR checklist

## Data availability
All data generated or analysed during this study are included in the manuscript and supporting files; source data files have been provided for all Figures.

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
