## [Editor Report · eLife Assessment]

This **important** study shows how genetic variation is associated with fecundity following a period of reproductive diapause in female *Drosophila*. The work identifies the olfactory system as central to successful diapause with associated changes in longevity and fecundity. While the methods used are **convincing**, a limitation of the study, as of any other laboratory-based investigation is the challenge of demonstrating how well measures for fitness related to diapause and its recovery correlates with realities encountered during development in the wild.

---

## [Referee Report · Reviewer #1 (Public review)]

Summary:

The paper begins with phenotyping the DGRP for post-diapause fecundity, which is used to map genes and variants associated with fecundity. There are overlaps with genes mapped in other studies and also functional enrichment of pathways including most surprisingly neuronal pathways. This somewhat explains the strong overlap with traits such as olfactory behaviors and circadian rhythm. The authors then go on to test genes by knocking them down effectively at 10 degrees. Two genes, Dip-gamma and sbb are identified as significantly associated with post-diapause fecundity, which they also find the effects to be specific to neurons. They further show that the neurons in the antenna but not arista are required for the effects of Dip-gamma and sbb. They show that removing antenna has a diapause specific lifespan extending effect, which is quite interesting. Finally, ionotropic receptor neurons are shown to be required for the diapause associated effects.

Strengths:

Overall I find the experiments rigorously done and interpretations sound. I have no further suggestions except an ANOVA to estimate heritability of the post-diapause fecundity trait, which is routinely done in the DGRP and offers a global parameter regarding how reliable phenotyping is.

Weaknesses:

A minor point is I cannot find how many DGRP lines are used.

---

## [Author Response]

The following is the authors’ response to the previous reviews.

**Reviewer #1 (Public Review):**
Summary:The paper begins with phenotyping the DGRP for post-diapause fecundity, which is used to map genes and variants associated with fecundity. There are overlaps with genes mapped in other studies and also functional enrichment of pathways including most surprisingly neuronal pathways. This somewhat explains the strong overlap with traits such as olfactory behaviors and circadian rhythm. The authors then go on to test genes by knocking them down effectively at 10 degrees. Two genes, Dip-gamma and sbb, are identified as significantly associated with post-diapause fecundity, and they also find the effects to be specific to neurons. They further show that the neurons in the antenna but not the arista are required for the effects of Dip-gamma and sbb. They show that removing the antenna has a diapause-specific lifespan-extending effect, which is quite interesting. Finally, ionotropic receptor neurons are shown to be required for the diapause-associated effects.Strengths and Weaknesses:Overall I find the experiments rigorously done and interpretations sound. I have no further suggestions except an ANOVA to estimate the heritability of the post-diapause fecundity trait, which is routinely done in the DGRP and offers a global parameter regarding how reliable phenotyping is.

We added to the Methods: “We performed a one-way ANOVA to get the mean squares for between-group and withingroup variances and calculated broad-sense heritability using the formula: *H2* = *MSG* - *MSE* / *MSG* + *(k-1) MSE* where *MSG* - Mean square between groups and *MSG* - Mean square within groups and k - Number of individuals per group. Using this formula, the broad-sense heritability for normalized post-diapause fecundity was found to be 0.51.”

We added to the Results: “The broad-sense heritability for normalized post-diapause fecundity was found to be 0.51 (see Methods).”

A minor point is I cannot find how many DGRP lines are used.

Response: We screened 193 lines and have added that to the Results.

**Reviewer #2 (Public Review):**
SummaryIn this study, Easwaran and Montell investigated the molecular, cellular, and genetic basis of adult reproductive diapause in *Drosophila* using the *Drosophila* Genetic Reference Panel (DGRP). Their GWAS revealed genes associated with variation in post-diapause fecundity across the DGRP and performed RNAi screens on these candidate genes. They also analyzed the functional implications of these genes, highlighting the role of genes involved in neural and germline development. In addition, in conjunction with other GWAS results, they noted the importance of the olfactory system within the nervous system, which was supported by genetic experiments. Overall, their solid research uncovered new aspects of adult diapause regulation and provided a useful reference for future studies in this field.Strengths:The authors used whole-genome sequenced DGRP to identify genes and regulatory mechanisms involved in adult diapause. The first *Drosophila* GWAS of diapause successfully uncovered many QTL underlying post-diapause fecundity variations across DGRP lines. Gene network analysis and comparative GWAS led them to reveal a key role for the olfactory system in diapause lifespan extension and post-diapause fecundity.Comments on revised version:While the authors have addressed many of the minor concerns raised by the reviewers, they have not fully resolved some of the key criticisms. Notably, two reviewers highlighted significant concerns regarding the phenotype and assay of post-diapause fecundity, which are critical to the study. The authors acknowledged that this assay could be confounded by the 'cold temperature endurance phenotype,' potentially altering the interpretation of their results.However, they responded by stating that it is not obvious how to separate these effects experimentally. This leaves the analysis in this research ambiguous, as also noted by Reviewer #3.

We should have clarified earlier that we actually chose to measure post-diapause fecundity in order to minimize any impact of ‘cold temperature endurance.” In fact, we chose post-diapause fecundity as the appropriate measure of successful diapause for both technical and conceptual reasons. Conceptually, the benefit of diapause is to perpetuate the species. It seems obvious to us that post-diapause fecundity is more relevant to species propagation than other measures of diapause such as how many egg chambers contain yolk or how many eggs are laid. Technically, we chose 5-week diapause and recovery based on pilot studies that showed that nearly all DGRP lines showed excellent survival at 5 weeks in diapause conditions. Therefore, our experimental design minimized as much as possible any effect of cold temperature endurance - in the sense of the ability to survive at 10°C - on our phenotype.

We apologize for not clarifying that point earlier and have added this text to the Results: “We chose 5 weeks based on pilot studies that showed that nearly all DGRP lines showed excellent survival at 5 weeks in diapause conditions while exhibiting sufficient variation in post-diapause fecundity to carry out GWAS. Beyond 5 weeks, fecundity was low and there was insufficient variation to conduct a GWAS.”

Additionally, I raised concerns about the validity of prioritizing genes with multiple associated variants. Although the authors agreed with this point, they did not revise the manuscript accordingly. The statement that 'Genes with multiple SNPs are good candidates for influencing diapause traits' is not a valid argument within the context of population and quantitative genetics.

We apologize for neglecting to revise the manuscript accordingly. We have revised Supplemental Table: S4 and ranked the genes by p-value.